# SkipPipe: Partial and Reordered Pipelining Framework for Training LLMs in Heterogeneous Networks

## Abstract

Data and pipeline parallelism are ubiquitous for training of Large Language Models (LLM) on distributed nodes. The need for cost-effective training has lead recent work to explore efficient communication arrangement for end to end training. Motivated by LLM's resistance to layer skipping and layer reordering, in this paper we explore stage (several consecutive layers) skipping in pipeline training, and challenge the conventional practice of sequential pipeline execution. We derive convergence and throughput constraints (guidelines) for pipelining with skipping and swapping pipeline stages. Based on these constraints, we propose `SkipPipe`, the first partial pipeline framework to reduce the end-to-end training time for LLMs with negligible effect on convergence, which we verify analytically and empirically. The core of `SkipPipe` is a path scheduling algorithm that optimizes the paths for individual microbatches and reduces their end-to-end execution time, complying with the given stage skipping ratio. We extensively evaluate `SkipPipe` on LLaMa models from 500M to 1.5B parameters on up to 20 nodes, through emulation and deployment prototypes. Our results show that `SkipPipe` reduces training iteration time by up to 50% compared to full pipeline. Additionally, our partial pipeline training also improves resistance to layer omission during inference, experiencing a drop in perplexity of only 2% when running only 75% of the model. Our code is available at `https://anonymous.4open.science/r/skippipe-43B2/`.

## 1 Introduction

Deep transformer-based architectures (Vaswani et al., 2017) have recently enabled unprecedented performance on complex language and cognitive tasks (Radford et al., 2018). These leaps can be explained by the ever growing corpora of available data and by the increasing size of (Large) Language Models (LLMs) (Touvron et al., 2023; Brown et al., 2020; Shoeybi et al., 2019). As a consequence, models are now too large to fit and be efficiently trained on a single GPU.

Distributed training techniques, such as Pipeline Parallelism (PP) and Data Parallelism (DP), become indispensable to efficiently train large models on distributed nodes (devices,GPUs). In the former the model is split in stages, containing non-overlapping sections of the model, across a set of nodes, which communicate sequentially between each other to run the whole model, thus forming a pipeline. In the latter, multiple pipelines train the model independently on different data batches, communicating between each other to synchronize the model weights after an update. Training with the standard synchronous algorithms and renting private clusters to train models can easily cost more than tens of thousands of dollars (Yuan et al., 2022), even for smaller models. Some prior work has proposed training on smaller clusters over a heterogeneous network (different communication latency and bandwidth between nodes), however in such a setting the communication between the GPUs is still one of the main limiting factors (Yuan et al., 2022).

Recent work has aimed to improve cost effectiveness of LLM training via heterogeneity-aware arrangement of the nodes (Yuan et al., 2022; Park et al., 2020; Um et al., 2024; Yan et al., 2024). Such methods present efficient arrangement of the GPUs to minimize the communication overhead in heterogeneous network settings. Yet, pipelining is done strictly following a sequential execution

Figure 1: An example of partial pipeline parallelism scheduling where each colored (solid or dashed) path represents a different microbatch. Each node in stage 0 sends out 2 microbatches, the first in solid, the second in dashed. Green backgrounds show the forward pass, while orange - the backwards pass. Arrows show the prioritisation of the microbatches from forward to backward pass within the same node. An example of a **collision** can be seen on node 7 during the forward pass, which subsequently delays the execution of the solid blue microbatch because of the dashed yellow microbatch.

of layers from beginning to the end for all microbatches (Huang et al., 2019; Qi et al., 2024; Yuan et al., 2022). The works of Bhojanapalli et al. (2021); Fan et al. (2020); Elhoushi et al. (2024) have demonstrated transformer architectures' robustness against layer skipping and even layer reordering during training and inference. We leverage this fact to propose a novel optimisation to traditional training - `SkipPipe`, which is the first partial pipeline framework that skips and re-orders pipeline stages. `SkipPipe` improves the training of the model on distributed nodes, with negligible degradation in performance, and is also suitable for communication heterogeneous settings. Moreover, the partial training via stage skipping in `SkipPipe` also improves the inference with layer/stage skips, which is beneficial for fault tolerant inference and early-exit strategies.

To minimize the end-to-end training latency via stage skipping and reordering, `SkipPipe` is composed of two modules: arranging nodes in stages, and a path scheduler for microbatches. For a given (heterogeneous) network of nodes and pipeline stages of an LLM model, `SkipPipe` allocates nodes to stages, where nodes in the same stage communicate in DP manner and nodes in different stages communicate in PP. Then, differently from standard pipelining where each microbatch passes through all stages sequentially along the same path, `SkipPipe` schedules partial paths for each microbatch that skip some stages and/or runs others out of order. As illustrated in Figure 1, each microbatch skips $k\%$ of the model where $k$ is a user-defined parameter.

The key challenge is how to select the path such that the number of microbatch collisions is minimised and the model convergence is not affected negatively. Our contributions can be summarised as follows: (i) We propose a novel and effective partial and reordered pipelining framework for distributed LLM training to reduce the communication overhead. (ii) We design a pipeline execution scheduler optimising the throughput for heterogenous network of nodes by utilising skipping and swapping stages and reducing collisions (overlapping microbatches executions). (iii) We evaluate our scheduler and present up to 250% reduction in iteration time when training with `SkipPipe` compared to training with a standard full-model framework in both emulated and real geo-distributed networks. Also, we demonstrate that there is minimal convergence degradation. (iv) We show that the models trained with `SkipPipe` also provide significant resistance to layer omission during inference, with a perplexity drop of only 2% when skipping a quarter the model.

## 2 SYSTEM SETTING

**System setup.** There are $\mathcal{N}$ distributed nodes for training an LLM model of $\mathcal{L}$ layers, which is divided in pipeline stages $\mathcal{S} := (S_0, S_1, \ldots, S_s)$. Each stage $S_i$ holds an (equal)[1] number of consecutive layers $L_j \ldots L_{j+\delta}$ and there are no overlapping layers across stages.

We assume each node has the same memory capacity allowing them to operate the same number of microbatches. Each node can communicate with any other and the communication cost between nodes is modelled with $(\mathcal{B}, \Lambda)$ matrices where communication between nodes $N_i$ and $N_j$ has a cost modelled by the latency $\lambda_{i,j} \in \Lambda$ and bandwidth $\beta_{i,j} \in \mathcal{B}$. Thus for a message of size $|msg|$, its

---

[1]Not necessary for our solution, but for simplicity and clarity we focus on the homogeneous stage/node case.

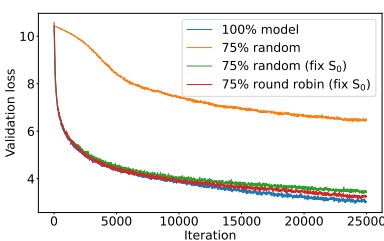

(a) Impact of skipped layer selection.

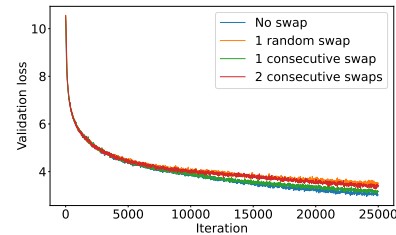

(b) Impact of stage swapping on full model.

Figure 2: Convergence of LLaMa-30M model. The validation loss is calculated for the whole model for every 50th iteration.

communication takes $\lambda_{i,j} + \frac{|msg|}{\beta_{i,j}}$ seconds. While communication may not be symmetric, since each link is used twice, once during forward and once during backward, we model latencies and bandwidth as the average of the two directions (e.g., $\lambda'_{i,j} = \frac{\lambda_{i,j}+\lambda_{j,i}}{2}$), as in Yuan et al. (2022).

**Distributed Training.** Each node is mapped to a single stage. To train the LLM with data and pipeline parallelism, a batch is split into microbatches. Nodes sharing the same stage communicate the gradient updates in DP, and nodes in different stages communicate activations in PP. We consider synchronous updates in pipelining - the weight update of an iteration is done after all the corresponding microbatches are processed. However, unlike common pipelining where each microbatch passes through all stages in the sequential order, we propose partial and reordered pipelining.

**Partial and Reordered Pipeline.** The prior work pinpointed that transformer-based architectures are robust to layer skipping, i.e., not executing a given layer (Bhojanapalli et al., 2021; Fan et al., 2020). We term skipping layers (or stages) in distributed training - partial pipeline parallelism. In the full pipeline scenario, microbatches traverse through the stages sequentially, e.g. $\mathcal{S} := (S_0, S_1, S_2, S_3, S_4, S_5)$. In our case microbatches can traverse through different sequences of stages, due to skipping a given stage ($\mathcal{S} := (S_0, S_1, S_4, S_5)$) or swapping the order of two stages ($\mathcal{S} := (S_0, S_1, S_3, S_2)$). The key research question is thus which stages should each microbatch run through, such that training time is minimized.

## 3 SKIPPIPE

In this section we present a novel approach to pipeline parallelism, employing skipping and swapping to reduce the required resources and increase throughput without degrading the training performance of LLMs. The goal is to find a viable partial pipeline schedule (paths of the microbatches) that minimizes the overall training latency given the number of microbatches target.

**Partial pipeline schedule** Given a DP and PP arrangement of nodes (a graph) with communication and computation limitations per link and node respectively, we find paths $p_1, p_2... \in \mathcal{P}$ (a sequence of nodes) for each microbatch such that end-to-end time to execute all microbatches is minimized. End-to-end time is the time for a single iteration between two data parallel rounds, including all PP computations and communications. Each path $p_i$ travels a sequence of nodes from a starting node back to itself (constituting forward and backward passes) where only $k\%$ of stages are skipped (and no stages are repeated in the path). The ordering of nodes in the backward pass needs to be the same as in the forward one. A path $p_i$ can be represented with respect to the stages ($p_i := S_{i_1}, \ldots, S_{i_l}$) or the nodes ($p_i := N_{i_1}, \ldots, N_{i_l}$) that it passes through, where $l := (100 - k)\%$ of the stages.

### 3.1 GUIDELINE FOR PARTIAL PIPELINING SCHEDULER

Here, we explain our guideline for a partial pipeline scheduler that selects the paths for each microbatch through a motivation example. We present three convergence and two throughput constraints to optimize the path selection. We derive the convergence constraints from our experimental results and previous work and the throughput constraints are based on the hardware and network limitations.

**Convergence Constraints**. To study the effects of stage skipping and swapping on the LLM convergence, we train a LLaMa-30M model (12 layers) divided in 6 stages with 2 layers each on the TinyStories dataset with 5 microbatches of size of 32 samples in two sets of experiments. In Figure 2 (a), we vary the selection of which stage to skip (for skipping percentage of 25%): random, random with no skipping the first stage, and round robin with no skipping the first stage (skip each intermediate stage equal number of times). By comparing the two random cases, we observe that the first stage is more critical than other stages and should not be skipped. Similar effect is also observed for larger transformer architectures (Bhojanapalli et al., 2021; Fan et al., 2020) and architectures with residual connections (Veit et al., 2016). Additionally, when we compare the random and round-robin cases, we see that convergence is better when each intermediate stage is skipped uniformly and trained for an equal number of microbatches. Figure 2 (b) shows that swapping execution order of two consecutive stages has negligible effect on the training loss, and swapping multiple stages or stages that are not consecutive causes more degradation. Using these observations, we derive the following **C**onvergence **C**onstraints for our path selection:

- `CC1`: A path $p_i$ never skips the first stage, i.e., $S_{i_1} = S_0 \ \forall p_i \in \mathcal{P}$.
- `CC2`: A path $p_i$ may run out of order at most two consecutive stages (1 swap), i.e., for a path $p_i = S_{i_1}, \ldots, S_{i_l}, |i_j - i_{j+1}| \leq 1 \ \forall j \in (1, l)$.
- `CC3`: Each stage $S_i \ (i \geq 1)$ is skipped for an equal amount of paths.

**Throughput Constraints.** In standard pipeline training, the whole model is executed sequentially and each node needs to receive activations of the microbatches from only one other node (the one before it/after it) in the forward pass/backward pass. In other words, each node receives only one microbatch to process at a time from each direction. However, as we introduce skips (and potentially swaps) in execution, it is possible for a node to simultaneously receive two microbatches from two different stages in the same direction, thus forcing the node to delay one of the microbatches. We refer to such cases as **collisions**, which can significantly degrade the end-to-end latency of a batch. To avoid collisions, we employ swaps to run stages out of order for a microbatch, thus utilising a currently idle node to reduce instantaneous overutilisation of another.

In addition, because of the caching of the activations that is needed for the backward pass, the number of active microbatches going through each node is limited by the memory of a node and denoted by $(m)$. Overall, we impose two **T**hroughput **C**onstraints:

- `TC1`: At most $m$ paths can go through each node $N_i$.
- `TC2`: Minimize collisions by swapping the pipelining order.

**Problem Formulation.** We formalise the optimization problem of partial pipeline scheduler as follows: For a given network of $\mathcal{N}$ nodes with bandwidth and latency matrices $(\mathcal{B}, \Lambda)$ and an LLM model consisting of pipeline stages $\mathcal{S}$, the number of microbatches $M$ and limitation of active microbatches $m$ per device, the partial pipeline scheduler aims to find the paths $\mathcal{P}$ that minimizes the maximum end-to-end latency across all microbatches of a given iteration:

$$\mathcal{P} \leftarrow \text{Scheduler}(\mathcal{N}, \mathcal{B}, \Lambda, \mathcal{S}, M, m)$$

such that $\mathcal{P} := \underset{\mathcal{P} \in \mathcal{P}_{ALL}, \ \forall p_i \in \mathcal{P}}{\arg \min \max} E2E(p_i)$ with constraints `CC1`, `CC2`, `CC3`, `TC1` and `TC2`

where $E2E(\cdot)$ is the end-to-end latency of a microbatch where the starting time of a microbatch is also taken into account, and $\mathcal{P}_{ALL}$ is the set of all possible sets of paths. Forming the paths is itself an NP-hard problem (as detailed in Section 3.3). We thus split the problem into two parts: first allocation nodes in stages under a full pipeline schedule and then finding the partial pipeline schedule for microbatches under the given node-to-stage mapping.

### 3.2 ALLOCATING NODES TO STAGES

For a given network of $\mathcal{N}$ nodes, cost matrices $(\mathcal{B}, \Lambda)$, and the pipeline stages $\mathcal{S}$, the initial node arrangement matches each node with a stage for standard full and sequential pipelining (no skips or swaps). This problem is already analyzed for heterogeneous networks in DT-FM (Yuan et al., 2022), solved through a two-phase optimiser: clustering of nodes for DP and then arrangement of the connections for PP. DP clustering can be seen as *graph partition problem* where each cluster

corresponds to a stage and the partition cost is bounded by the slowest communication between two nodes in the same stage. Then these clusters are ordered for PP, which can be represented as an *open-loop Traveling Salesman problem* (Papadimitriou, 1977). This problem can be solved via genetic algorithms as described in Yuan et al. (2022).

To allocate nodes to stages in `SkipPipe`, we modify the algorithm given in DT-FM for the un-balanced cluster sizes. Following convergence constraints `CC1` and `CC3`, the initial stage is never skipped whereas all other stages are skipped equally, so that $k\%$ of the stages are skipped for each microbatch. Assuming the nodes allocated in a stage is $S_i(\mathbf{n})$, we formulate the number of nodes per stages with the following equation:

$$|S_i(\mathbf{n})| = |S_0(\mathbf{n})| \left(1 - \frac{s}{s-1} \cdot \frac{k}{100}\right) \ \forall i \in (1, s). \tag{1}$$

To balance the workload across stages, we allocate the nodes per stage using the ratio given above. Thus, unlike DT-FM setting, we require more nodes in the first stage. With the optimised arrangement of nodes in stages, we can look for paths through the system that would satisfy our constraints.

### 3.3 Partial and Reordered Pipelining

Once nodes are arranged into stages, we schedule the microbatches through the system by skipping and reordering stages, which is the core of `SkipPipe`. It is important to note the difference between a path and a microbatch. While a microbatch does travel down a path, multiple microbatches may use the same path. For example, when a node completes a backwards pass for a given microbatch, it can reuse the path it had just traversed, as it is the one that immediately has nodes with free memory. Thus we find a set of paths for the first wave of microbatches and reuse them a number of times during an iteration to meet the desired batch size.

Given our formulation, we model the problem as a *continuous-time Multi Agent Path Finding* (MAPF) problem (Andreychuk et al., 2021). In such problems a number of agents with some starting location must traverse a graph to reach their end goals. Thus, we reuse the graph of the node arrangement, where the cost on each edge is the time to communicate one microbatch. Each agent represents a microbatch which travels from a starting node in stage $S_0$ to the same destination node while passing $s(100 - k)/100$ nodes in total. An agent can either wait at a node, move through the node (computation), or move to a different node via the corresponding edge (communication). Each move is associated with a given cost. In the continuous-time setting, actions do not take 1 unit of time, but can be of arbitrary length. The problem has the additional constraints that no two agents can collide (be on the same node at the same time). Thus, due to the nodes' real physical limitations, we allow traversal of only one agent at a time through a node (constraint `TC2`). To find a viable solution we employ a modified version of the *continuous-time Conflict-Based Search* (CBS) (Andreychuk et al., 2021) based on the changes described above.

The first four constraints (`CC1`, `CC2`, `CC3`, `TC1`) are merely about finding the paths, while constraint `TC2` deals with conflicts between two agents. `CC1` and `CC2` are individual constraints per agent and thus can be solved by an A* search (Doran & Michie, 1966). We use A* (instead of the Safe interval path planning used in Andreychuk et al. (2021)) that allows us to model the skips, swaps, and the additional constraints better. However, `CC3` and `TC1` require inter-agent optimization as they specify global constraints - limiting the number of agents that can go through a node per iteration. This requires knowing all other agent's paths, making an A* solver insufficient. We thus delegate all constraints, apart from `CC1` and `CC2` to be resolved by CBS, with for `CC3` and `TC1` setting a constraint that an agent cannot visit all nodes in a stage or a specific node respectively, from $(-\inf, \inf)$. However, this proves extremely costly for large graphs or large number of agents, as an exponential number of possible solutions would need to be explored, before resolving `TC2` constraints. We thus approximate the optimal solution, by employing a heuristic idea: whenever possible we exclude the slowest agents of each (starting) node from adding constraints as any additional constraint would detrimentally affect the slowest path. First, we employ CBS to find a number of solutions that satisfy `CC1`, `CC2` and `CC3` constraints.[2] Then for these generated solutions we solve for `TC1` constraints. Once no `TC1` constraints are detected, `TC2` constraints are checked. A constraint `TC2` is

---

[2]We choose 32 solutions in our experiments, as this proved sufficient to find good solutions, without expanding the subsequent search space too much.

added for each relevant agent by specifying that they cannot visit the conflicting node for the duration the other agent is traversing it.

### 3.4 PATH FINDING

Here we describe our path finding algorithm, the detailed steps of which can be found in Appx. C in Algorithm 1. To find a set of paths satisfying the current constraints, we employ A* for each agent with a time dimension. When an agent travels between two nodes, its time is increased by the time it takes for a microbatch to travel down that link. Whenever an agent travels through a node, its time is increased by the time the node takes to process a microbatch. If an agent is to visit a node and during the processing time there is a constraint that prohibits the agent from being in that node, its time of visiting the node is delayed to the end of constraint.

An agent must skip exactly $k\%$ of the stages. Thus when expanding a node, we do not consider the starting node until this condition is met. When we visit again the starting node the time of a forward and a backward pass, given all constraints, is estimated and the node is readded to the heap with that cost and a special flag marking it as a potential final solution. When a node marked as a potential solution is popped from the heap, it is returned as the current fastest path for that agent that satisfies all current constraints.

Unlike traditional A* we do not make use of a visited set - we may consider a node during our search multiple times. This is because *how* we expand the starting node in the A* search, may not be the fastest way to do a forward + backward pass (which is why we re-add the starting node with the special flag). When expanding an A* node, we exclude from the set of potential next nodes all nodes that have been on that path or belong to a stage that has been visited. We may perform at most 1 swap in the ordering of stages for a given path (`CC2` constraint). Nodes that would go over this limit are excluded from consideration.

### 3.5 THEORETICAL ANALYSIS OF SKIPPIPE

We base our analysis by relating the problem to that of Stochastic Depth (Huang et al., 2016), as our method is similar to training in such manner with uniform survival chance per layer. Thus convergence proof of SkipPipe is equivalent of the work of Hayou & Ayed (2021) demonstrating that training with Stochastic Depth and survival chance of ($p_l := \frac{100-k}{100}$) with additional Gaussian noise per input acts as a regularizer. This is formalised in the following theorem where $\delta$ is a binary variable dictating if the given layer is used.

**Theorem 3.1.** *(Hayou & Ayed, 2021) For input $x$, let $y_i^j$ be the activations of the j-th neuron before the i-th layer, $z_l$ the activations after the l-th layer, layer l of N, $p_l = \frac{100-k}{100}$, $X_{l,N} = (\delta_l - p_l)\mu_{l,N}(x)$, with $l \leq i$, $\mu_{l,N}(x) = \langle z_l, \nabla_{y_l} y_l^j(x, \mathbf{1})\rangle$, and $Var_\delta[X_{l,N}(x)] = p_l(1-p_l)\mu_{l,N}(x)^2$, assume that:*

- *There exists $a \in (0, \frac{1}{2})$ such that for all N and $l \in N$, $p_l \in (a, 1-a)$,*

- *$lim_{N\to\infty} \frac{max_{k\in N}\mu_{k,N}(x)}{\sum_{l\in N}\mu_{l,N}(x)} = 0$, i.e. no single layer dominates in the computation,*

- *$v_{\frac{l}{N},\infty}(x) = lim_{N\to\infty} \frac{\sum_{l=1}^N Var_\delta[X_{l,N}(x)]}{N}$ exists and is finite.*

*Then, as $lim_{N\to\infty} y_l^j(x, \delta) \sim y_l^j(x, p) + \mathcal{N}(0, \frac{l}{N} v_{\frac{l}{N},\infty}(x))$*

The details of the proof are presented in Appendix F.

### 3.6 PERFORMANCE STABILIZATION

Based on the proof above, we can see that training with vanilla SkipPipe approximates training a partial model. To improve the performance of the model in full-execution scenarios, we also added occasional full-model training steps (i.e. steps where no skips or swaps are performed). Regarding the frequency of the full-model training step, empirically we find that performing such a step once every 10 training steps (so 9 with skips and 1 full) yields good convergence results, without sacrificing the throughput. In Appendix G.1 we present our experimental ablation study on this.

## 4 EXPERIMENTAL RESULTS OF SKIPPIPE

We demonstrate that `SkipPipe` provides significant improvement in the iteration throughput and provides faster convergence in terms of wall-clock time in geo-distributed settings. For all experiments we do 1 full iteration every 10 iterations. All schedulers are limited to 1 swap per microbatch. For throughput experiments, we investigate the speed up of our partial pipeline scheduler `SkipPipe` wrt. the baseline SOTA schedulers on a LLaMa-1.5B model. For convergence results, we demonstrate two types of training - pretraining from scratch (a LLaMa 500M model on the RedPajamas dataset (Weber et al., 2024)) and supervised fine-tuning (a LLaMa 3.2 1B model on the Tulu dataset (Lambert et al., 2024)), with different skip ratios. We observe that using `SkipPipe`, the models converge at the same rate (with negligible difference in performance) but with a significantly higher throughput, meaning that training converges much faster in wall-clock time.

### 4.1 THROUGHPUT

We evaluate the throughput improvement of `SkipPipe` by measuring the end to end time for pipeline training of an iteration. We test a LLaMa-1.5B model distributed training (see Appendix A for architecture details) with 3 different skipping ratios (0%, 25% and 33%) and different number of nodes. We analyze the throughput with both simulated environment where we can control the network delays and the real deployment. For the first case, we utilise H100 nodes and their communication is simulated by the bandwidth and latency values given in DT-FM (Yuan et al., 2022). For the real deployment, we rent T4 nodes on Google Cloud across 12 different locations and 5 continents.

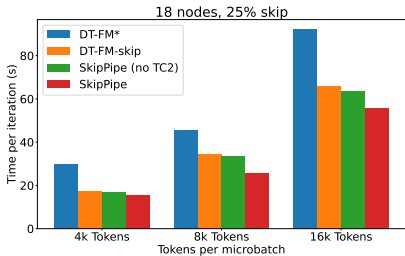

(a) Heterogeneous simulated setup with 18 nodes and 25% skip rate.

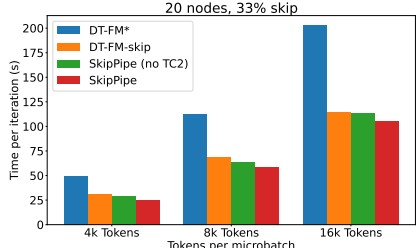

(b) Heterogeneous simulated setup with 20 nodes and 33% skip rate.

Figure 3: Time per iteration of four schedulers with two skip percentages (25% and 33%) and three token numbers (4K, 8K and 16K). DT-FM* representing the compensated results for the baseline with no skips, DT-FM-skip uses node arrangement of DT-FM and skips $k\%$ with additional constraints (see Appendix B.1), `SkipPipe` (no `TC2`) is our scheduler without `TC2`.

**Simulation** In Figure 3, we present the experimental results for two skip percentages ($k$ :=25% and 33%) and 4 different schedulers. We compare our scheduler, `SkipPipe`, with (1) DT-FM: 0% skip training using DT-FM scheduler, (2) DT-FM-skip: $k\%$ skip training using DT-FM scheduler with additional constraints (see Appendix B.1), (3) `SkipPipe` (no `TC2`): $k\%$ skip training using our scheduler `SkipPipe` where the collision constraint `TC2` is ignored. We test with varying number of microbatch sizes - of 1, 2 and 4, and use gradient accumulation for each. The time per iteration values are averaged over 50 iterations. Since we optimise the schedule for a given node/stage allocation, we measure the pipeline time and omit the data parallelism part where weight aggregation happens because the aggregation time is the same for a fixed node/stage allocation regardless of the microbatch paths. Finally, we perform one warm-up iteration where nodes discover each other.

In Figure 3a, we have the results for 25% skip case. We tested 4 stages with 18 nodes where the nodes are distributed to the stages according to Equation 1: $(6, 4, 4, 4)$, except the 0% skipping case used in DT-FM baseline. To keep the node/stage sizes the same, for the DT-FM baseline, we use 16 nodes where nodes are equally distributed $(4, 4, 4, 4)$. To (over)compensate the baseline case using less nodes, we project their performance by proportionally reducing the end-to-end latency. Specifically, we multiply the latency of baseline by $\frac{16}{18}$, and these compensated latency results are represented by DT-FM*. Note that considering the communication of those additional nodes being ignored, this is a

lower bound of their performance. Here, `SkipPipe` achieves $35 - 45\%$ improvement compared to the baseline in the 8K and 16K tokens case. In Figure 3b, we have the results for 33% skip case where we tested 6 stages with 20 nodes. Similarly to the above case, number of nodes per stage is $(5, 3, 3, 3, 3, 3)$, except the baseline, which is compensated by multiplying the corresponding latency values with $\frac{18}{20}$. We observe that removal of collisions (`TC2`) provides a speedup of 10%, and all constraints together yield to more than $45\%$ speedup compared to DT-FM*.

**Real Deployment** We test the throughput of `SkipPipe` in a *real* geo-distributed deployment across 12 locations (see Appendix E for detailed results). We observe much higher speed up in our deployment: **250%** speed up relative to a DT-FM baseline. We attribute this greater speed up due to the much higher variability in bandwidth between locations relative to the ones used in Yuan et al. (2022). Additionally, as `SkipPipe` has lower memory consumption due to the skips, it can process much higher number of microbatches in a single forward-backward wave. Thus, DT-FM needs more forward-backward waves to reach the same batch size, incurring much higher time per iteration. Furthermore, addition of skipping into DT-FM (DT-FM skip) does not optimize for collisions or skips, thus being around 20% slower compared to our solution.

## 4.2 CONVERGENCE

Here we show that our scheduler `SkipPipe` does not degrade the convergence of the training compared to the baseline. We verify this by training from scratch a LLaMa-500M on the RedPajamas data (Weber et al., 2024) and finetuning LLaMa-1B model (Dubey et al., 2024) on the Tulu 3 dataset (Lambert et al., 2024) with three different skip rates - 0% (baseline), 25%, and 33% skips.

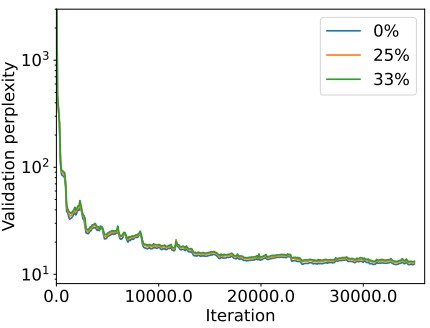

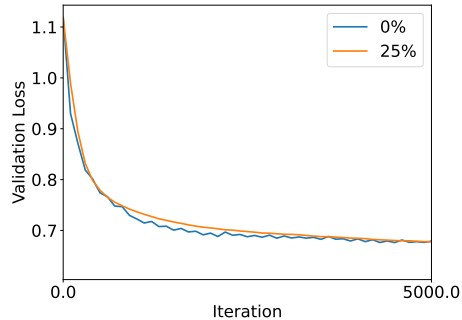

(a) Training LLaMa-500M with RedPajamas.

(b) Finetuning LLaMa-1B with Tulu 3 dataset.

Figure 4: Convergence of validation loss (of the full model) with $33\%$, $25\%$ and $0\%$ skip rates.

In Figure 4, we report the validation perplexity loss every 100th iteration by running the entire model (regardless of the training schedule). Our experiments show that `SkipPipe` achieves similar convergence to the baseline for both training (see Figure 4a) and finetuning (see Figure 4b), despite training with a fraction of the model each time. Also, since `SkipPipe` has a much higher throughput, convergence in terms of wall-clock time is significantly faster. We further evaluate the inference perplexity of the final models on a few common common datasets in the first column of Table 1. Similar to the convergence curves, we observe minimal negligible difference between the models trained with SkipPipe and those without. We present inference performance of the 500M models in the following section. Performance of the 1B finetuned models can be found in Appendix G.2.

## 4.3 FAULT TOLERANT INFERENCE

By training with `SkipPipe`, the models exhibit robust inference results even if some stages fail (except the first one). We demonstrate this by evaluating the trained Llama-500M models (in Section 4.2) for various inference stage skip rates on Wikipedia (Computer, 2023), Gutenberg (pro), and Stackexchange datasets (Computer, 2023). For each skip rate a corresponding number of stages is dropped at random per sample.

Table 1: Perplexity on several dataset across 1000 evaluation samples for various skip rates. The inference/training skip rate shows the percentage of stages being skipped during inference/training.

| Inf. skip rate | 0% | | | 25% | | | 33% | | | 50% | | |
|---|---|---|---|---|---|---|---|---|---|---|---|---|
| Training skip rate | 0% | 25% | 33% | 0% | 25% | 33% | 0% | 25% | 33% | 0% | 25% | 33% |
| **Wikipedia** ↓ | 18.8 | **18.7** | 19.1 | 75.4 | **19.1** | - | 115 | - | **22.6** | 457.6 | 36.1 | **28.7** |
| **Gutenberg** ↓ | **26.8** | 27 | 27.6 | 143.8 | **28.3** | - | 214.3 | - | **35.3** | 488.1 | 53.7 | **45.9** |
| **Stackexchange** ↓ | **29.5** | 29.7 | 30.3 | 110 | **31** | - | 189.2 | - | **38.4** | 560 | 58.9 | **48.4** |

As seen in Table 1, our partial pipelining provides robustness against arbitrary stage removal during inference time. Overall, we observe that for downstream task, our skip method provides similar performance as a baseline schedule, as supported by the convergence graphs. Interestingly, when executing only 75% of the model, our solution experiences less than 2% perplexity drop for models trained with a 25% skip schedule. As these models are not instruction finetuned, we leave more in-depth evaluation on downstream tasks as future work.

## 5 RELATED WORK

**Efficient and Heterogeneity-aware Distributed Training.** There have been several works to improve (communication) efficiency of LLM training (Douillard et al., 2023; Peng et al., 2024; Jaghouar et al., 2024) where they show that the communication overhead can be significantly reduced by minimizing the synchronization for gradients in DP. Moreover, there are several heterogeneity-aware scheduling methods (Yuan et al., 2022; Ryabinin et al., 2023; Park et al., 2020; Um et al., 2024; Yan et al., 2024) proposing efficient DP and PP arrangement of the nodes to minimize the communication overhead. Yet, pipelining is always done in a sequential execution of all layers (Huang et al., 2019; Qi et al., 2024; Harlap et al., 2018; Park et al., 2020). To the best of our knowledge, no prior work has studied the opportunity of optimizing for partial pipeline usage.

**Skip Connections and Early Exit.** Models employing skip connections have been known to exhibit robustness to random layer omission and perturbation (Veit et al., 2016; Bhojanapalli et al., 2021). Works such as Huang et al. (2016) demonstrated how larger models can be trained with less resources, by skipping certain layers during training. LayerDrop (Fan et al., 2020) demonstrated that models trained partially are more robust to layer omission during inference. Based on this work, Layerskip (Elhoushi et al., 2024) proposed a novel training approach and loss function, which enabled them to perform early exiting during inference - running only part of the model to generate tokens and using the whole model only to verify their probability.

## 6 CONCLUSION AND FUTURE WORK

Training LLMs requires a significant number of GPUs and enormous training data. The have been many works on communication and computation improvements for DP and PP methods aiming to achieve a cost-effective training. Yet, existing PP methods are limited to the sequential execution of the layers. In this paper we introduce a novel approach to partial and reordered pipeline parallelism, `SkipPipe`, which allows stage skips and swaps of stages. Our experiments showed `SkipPipe` achieves up to 50% throughput improvement without significantly affecting the convergence per iteration. Due to resource limitations, we have tested with LLaMa-like models up to 1.5B parameters, and we leave experiments with even larger models and different architectures as a future work. Moreover, our partial training also produces models resistant to layer removals during inference, which makes them suitable for early exit and fault tolerant inference. A LLaMa-500M model trained with `SkipPipe` experiences a drop in perplexity of only 2% when skipping a quarter of the model. As future work we aim to make use of this fault-tolerant inference for the purposes of early exiting. Finally, while this paper focuses on the homogeneous nodes/ heterogeneous network, in future work, we plan to extend our solution to the full heterogeneous setting where nodes can have different memory and computational capacities.

4

## 7 REPRODUCIBILITY STATEMENT

We keep the code for the proposed `SkipPipe` and baselines on the following open sourced repository anonymously: `https://anonymous.4open.science/r/skippipe-43B2/`. Our repository contains the necessary scripts to execute the full training of foundational models on distributed computing nodes, e.g., cloud nodes, from scratch.

## 8 ETHICAL STATEMENT

We conform to the ICLR code of ethics. Our work introduces an efficient manner of pretraining and finetuning foundational model in geo-distributed networks, thus democratising access to LLM training. We transparently report all findings and results to inform future research and applications.

We do not make use of LLMs for ideating or writing. LLMs were used for the purposes of this work to train models and evaluate their performance and training time.

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

## A    Model Configurations

We perform all our experiments with LLaMa-based Touvron et al. (2023) model architectures with the Sentence Piece Tokenizer Kudo & Richardson (2018). The different models and their parameters are shown in Table 2.

Table 2: Model parameters.

| Model | Dim | Heads | Layers | Context |
|---|---|---|---|---|
| LLaMa 50M | 288 | 6 | 12 | 256 |
| LLaMa 500M | 1024 | 16 | 24 | 1024 |
| LLaMa 1.5B Elhoushi et al. (2024) | 2048 | 16 | 24 | 4096 |
| LLaMa-3 1B Dubey et al. (2024) | 2048 | 32 | 16 | 8192 |

## B    Test Configurations

Configurations of the throughput tests are presented in Table 3. Stage sizes for the $33\%$ case are (5,3,3,3,3,3) (6 stages, 5 of size 3 and 1 of size 5), and for the $25\%$ case - (6,4,4,4) (4 stages, 3 of size 4, 1 of size 6).

For the convergence test, 8 samples per microbatch were used, with a total batch size of 500k tokens. Learning rate was set to $4 \times 10^{-4}$ and gradient norms were clipped to 1.0.

### B.1    DT-FM-skip path selection

Here we explain how the DT-FM-skip is determined. We choose paths that satisfy constraints `CC1` in an optimised arrangement of nodes in stages. DT-FM-skip serves as a skip baseline which is mainly optimised for the initial node arrangement, but not necessarily for the partial microbatch paths.

In order to keep comparison fair, we chose to satisfy constraints `TC1`, as otherwise delays will be introduced on nodes whose memory is exceeded, as it will need to wait for a backwards pass to come through, before it can continue with this forward pass. Due to this, and our experiment setups, we also inadvertently would satisfy constraints `CC3`. Thus the algorithm for determining the paths for this baseline is identical to that of the non-collision aware one, except that the computation time of each node and communication time between nodes is set to 1. Thus the algorithm does not optimise for fastest paths or `TC2` constraints.

## C    Detailed Path Selection Algorithm

In Algorithm 1 we present the steps of our path selection function.

## D    Possible Extensions of Our Algorithm

### D.1    Path Coarsening

Here we also present an alternative path finding method based on path coarsening that finds solutions faster, but they may be sub-optimal. The reason for the sub-optimality is that it may increase idle time on devices. However, in a strictly homogeneous device memory setting, it can ignore `TC2` constraints. Thus, it is best suited for large systems of nodes with equal memory capabilities, where an exact solution may be too costly to compute and due to the homogeneity of the system, most quality solutions will have similar throughput.

Here we make use of **path coarsening** - grouping multiple paths into one meta-agent. Meta-agents traverse a node sequentially, without interruption, and take the total amount of execution time of all

Table 3: Test settings.

| Skip | Path finding | World size | Samples per MB | Batch size (tokens) |
|------|-------------|-----------|---------------|--------------------|
| 0% | DTFM Yuan et al. (2022) | $18 \rightarrow 20^a$ | 1 | 184K |
| 0% | DTFM Yuan et al. (2022) | $18 \rightarrow 20^a$ | 2 | 368K |
| 0% | DTFM Yuan et al. (2022) | $18 \rightarrow 20^a$ | 4 | 737K |
| 33% | DT-FM-skip | 20 | 1 | 184K |
| 33% | DT-FM-skip | 20 | 2 | 368K |
| 33% | DT-FM-skip | 20 | 4 | 737K |
| 33% | non-collision aware | 20 | 1 | 184K |
| 33% | non-collision aware | 20 | 2 | 368K |
| 33% | non-collision aware | 20 | 4 | 737K |
| 33% | collision aware | 20 | 1 | 184K |
| 33% | collision aware | 20 | 2 | 368K |
| 33% | collision aware | 20 | 4 | 737K |
| 0% | DTFM Yuan et al. (2022) | $16 \rightarrow 18^b$ | 1 | 147K |
| 0% | DTFM Yuan et al. (2022) | $16 \rightarrow 18^b$ | 2 | 294K |
| 0% | DTFM Yuan et al. (2022) | $16 \rightarrow 18^b$ | 4 | 589K |
| 25% | DT-FM-skip | 18 | 1 | 147K |
| 25% | DT-FM-skip | 18 | 2 | 294K |
| 25% | DT-FM-skip | 18 | 4 | 589K |
| 25% | non-collision aware | 18 | 1 | 147K |
| 25% | non-collision aware | 18 | 2 | 294K |
| 25% | non-collision aware | 18 | 4 | 589K |
| 25% | collision aware | 18 | 1 | 147K |
| 25% | collision aware | 18 | 2 | 294K |
| 25% | collision aware | 18 | 4 | 589K |

[a] In 33% skip experiment, we use 6 stages with $(5, 3, 3, 3, 3, 3)$ nodes. DT-FM 0% skip does not use extra nodes in the first stage (as all stages are used equally). To (over)compensate them using less nodes (while keeping the stage sizes the same), we project their performance by linearly reducing the latency accordingly. In other words, if an iteration of DT-FM case takes 20sn with 18 nodes, we assume it would take 18sn with 20 nodes. Considering the communication of those additional nodes being ignored, this is upper bound of their performance.

[b] Same with above except in 25% skip experiment, we use 4 stages with $(6, 4, 4, 4)$ nodes. Therefore, 16 nodes are projected to 18 nodes.

---

**Algorithm 1** Path Selection Function.

---

**Require:** $\mathcal{S}, k\%, G$ - initial node/stage arrangement
**Ensure:** $\mathcal{P}$
  1: $\mathcal{O} \leftarrow \emptyset$
  2: $T_{constraints} \leftarrow \emptyset$
  3: Assign $S_0$ to the first stage of $T_{paths}$
  4: $T_{paths} \leftarrow$ find paths via A*($G, T_{constraints}$)
  5: Order $T_{paths}$ by their time to complete in ascending order
  6: $T_{cost} \leftarrow$ time for slowest agent to complete route
  7: Insert $T$ into $Open$
  8: **while** $|\mathcal{O}| < 32$ **do**
  9:     $T \leftarrow$ best solution from $Open$
 10:     Check for $S_i$ in $T$ which has more than $|\mathcal{S}|k\%$ agents going through other than $S_0$
 11:     **if** conflict **then**
 12:         $\mathcal{K} \leftarrow$ number of agents going through $S_i$
 13:         $Solution \leftarrow$ new node
 14:         $Solution_{constraints} \leftarrow T_{constraints}$
 15:         **for** each $\mathcal{D}_m \in S_i$ **do**
 16:             **for** each of the $\mathcal{K} - |\mathcal{P}|k\%$ fastest paths $p \in \mathcal{P}$ going through $S_i$ **do**
 17:                 $Solution_{constraints} \leftarrow Solution_{constraints} + (p, -\inf, \inf, \mathcal{D}_m)$
 18:             **end for**
 19:         **end for**
 20:         $Solution_{paths} \leftarrow$ find paths via A*($G, Solution_{constraints}$)
 21:         Order $Solution_{paths}$ by their time to complete in ascending order
 22:         $Solution_{cost} \leftarrow$ time for slowest agent to complete route
 23:         Insert $Solution$ into $Open$
 24:     **else**
 25:         $\mathcal{O} \leftarrow \mathcal{O} \cup T$
 26:     **end if**
 27: **end while**
 28: **while** $\mathcal{O}$ is not empty **do**
 29:     $T \leftarrow$ best solution from $\mathcal{O}$
 30:     Check for conflicts TC1 or TC2 in $T$
 31:     **if** conflict of type TC1 **then**
 32:         $D_k$ would be the node, whose $m$ is exceeded as per TC1
 33:         $\mathcal{K}$ the paths that go through $D_m$
 34:         $Solution \leftarrow$ new node
 35:         **for** each of the $\mathcal{K} - m$ fastest paths $p \in \mathcal{P}$ going through $D_k$ **do**
 36:             $Solution_{constraints} \leftarrow Solution_{constraints} + (p, -\inf, \inf, \mathcal{D}_k)$
 37:         **end for**
 38:         $Solution_{paths} \leftarrow$ find paths via A*($G, Solution_{constraints}$)
 39:         Order $Solution_{paths}$ by their time to complete in ascending order
 40:         $Solution_{cost} \leftarrow$ time for slowest agent to complete route
 41:         Insert $Solution$ into $\mathcal{O}$
 42:     **else if** conflict of type TC2 **then**
 43:         Two paths, $p_i$ and $p_j$ collide on $D_k$. Each of them is at the node during the intervals $t_{s,i}, t_{e,i}$ and $t_{s,j}, t_{e,j}$, respectively
 44:         $Solution \leftarrow$ new node
 45:         **if** $E2E(p_i) > E2E(p_j)$ or $|E2E(p_j) - E2E(p_j)| < \delta$ **then**
 46:             $Solution_{constraints} \leftarrow T_{constraints} + (p_j, t_{s,i}, t_{e,i}, D_k)$
 47:         **end if**
 48:         **if** $E2E(p_i) < E2E(p_j)$ or $|E2E(p_j) - E2E(p_j)| < \delta$ **then**
 49:             $Solution_{constraints} \leftarrow T_{constraints} + (p_i, t_{s,j}, t_{e,j}, D_k)$
 50:         **end if**
 51:         $Solution_{paths} \leftarrow$ find paths via A*($G, Solution_{constraints}$)
 52:         Order $Solution_{paths}$ by their time to complete in ascending order
 53:         $Solution_{cost} \leftarrow$ time for slowest agent to complete route
 54:         Insert $Solution$ into $\mathcal{O}$
 55:     **else**
 56:         **Return** $\mathcal{P} \leftarrow T_{paths}$
 57:     **end if**
 58: **end while**
 59: **Return** $\emptyset$

---

the microbatches in the meta-agent. Meta-agents thus become 2-dimensional objects, rather than the point-agents we were considering prior. The downside is that in heterogeneous environments, meta-agents might become more stretched out or mode condensed as they traverse the system. Consider three nodes arranged as A-B-C, taking time to process a microbatch of respectively 1, 2, and 1 seconds. communication between them is 1 second per microbatch. Initially, a meta-agent of 2 microbatches, would have a size of 2 seconds at node A. At node B, due to its delay of processing, the agent will be resized to size of 4, even though the subsequent node would have a gap of 1 second where it would be idle between the two microbatches. However, with meta-agents with multiple paths this level of detail is lost in favour of faster solutions. The best benefit of path coarsening is in a fully homogeneous node setting - equal processing time and equal memory for each. In such a setting we can create meta-agents with number of microbatches in them equal to the memory of the nodes. When finding the solution, all meta-agents will have mutually exclusive paths, thus no collisions need to be considered. Proving the optimality of such a solution is beyond the scope of the paper.

In fact our solution has already made use of a degree of coarsening, as we optimise only the first forward pass in an iteration. It is possible to find an even better solution across where no path is reused by microbatches, however, due to the difficulty of finding such a solution even for a small world and small number of agents, we have not performed further analysis.

### D.2   MULTIPLE SWAPS

It is possible to increase the number of swaps by introducing some linear penalty for paths that have swaps more than the desired amount, as a higher number of swaps hampers convergence, but may increase throughput. It is also possible to define an additional constraint that sets a maximum number of swaps across all paths, which would be delegated to CBS to resolve like constraint `CC3`, e.g. at most $|\mathcal{P}|$ swaps across all paths. This would however greatly increase the time to find a quality solution.

## E   GOOGLE CLOUD INTER-LOCATION TESTS

In this section we repeat the 33% experiment in Section 4.1, however instead of simulating the delays and bandwidths, we rent 20 T4 nodes on Google Cloud across 12 different locations and 5 different continents. The bandwidth between the locations is provided in Fig. 6. These were measured between locations for 5 minutes of traffic. Bandwidth is given in GB/s.

We present our findings in Fig. 5. Compared to the DT-FM baseline, SkipPipe achieves almost **250%** speed up, which is much higher than our simulation results. We attribute this greater speed up due to the much higher variability in bandwidth between locations relative to the ones used in Yuan et al. (2022). Additionally, as SkipPipe has lower memory consumption due to the skips, it can process much higher number of microbatches in a single forward-backward wave. Thus, DT-FM needs more forward-backward waves to reach the same batch size. In comparison to DT-FM skip and SkipPipe (no TC2), we see similar results as in our simulated ones where SkipPipe is 10-15% faster than with no TC2 case and 35% faster than DT-FM skip case.

## F   CONVERGENCE PROOF

The theorem and proof are almost verbatim replicas of the ones given in Hayou & Ayed (2021). Here, we repeat them with our notation for the sake of completeness.

We define a residual model of N layers as $\mathbf{W}_N = (I + \delta_n F_n).. \circ (I + \delta_1 F_1)$ where each $\delta$ is either 0 or 1, thus describing whether the given layer is used or not. The $\delta^*$ vector terms all $\delta$ values, i.e. the mask describing which layer/stage is used. A mask of $\mathbf{1}$ would mean that every layer is used.

For input $x$, let $y_l^j$ be the activations of the $j$-th neuron before the $l$-th layer, and $z_l$ be the activation output after the $l$-th layer. A neuron's pre-activations can be approximated via first order Taylor expansion around $\delta^* = 1$ as:

$y_i^j(x, \delta^*) \approx y_i^j(x, 1) + \frac{1}{N} \sum_{l=1}^i (p_l - 1)\langle z_l, \nabla_{y_l} y_l^j(x, \mathbf{1})\rangle + \frac{1}{\sqrt{N}} \sum_{l=1}^i (\delta_l - p_l)\langle z_l, \nabla_{y_l} y_l^j(x, \mathbf{1})\rangle$. Let's term $\mu_{l,N}(x) = \langle z_l, \nabla_{y_l} y_l^j(x, \mathbf{1})\rangle$, $X_{l,N}^1(x) = (p_l - 1)\mu_{l,N}(x)$ and $X_{l,N}^2(x) = (\delta_l - p_l)\mu_{l,N}(x)$.

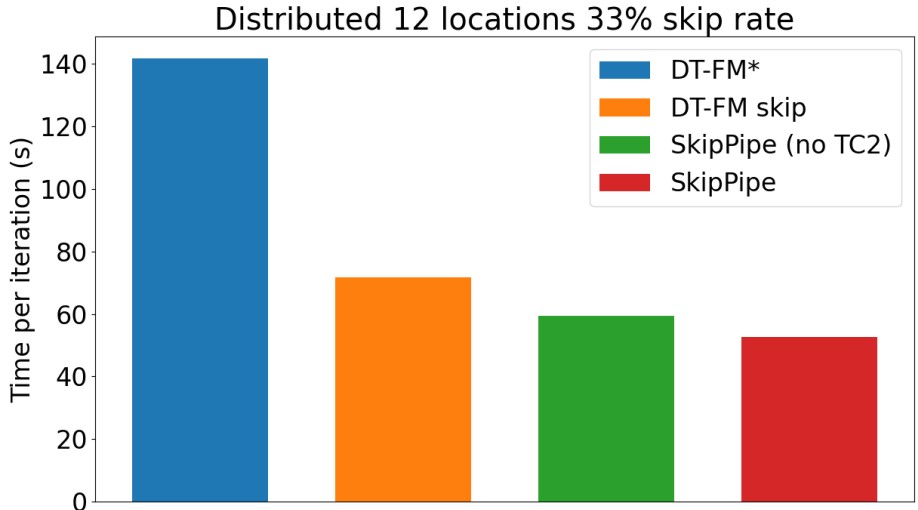

Figure 5: Real deployment with 20 nodes and 33% skip rate (8k tokens).

**Theorem F.1.** *Hayou & Ayed (2021) For input $x$, let $y_i^j$ be the activations of the j-th neuron before the i-th layer, $z_l$ the activations after the l-th layer, layer l of N, $p_l = \frac{100-k}{100}$, $X_{l,N} = (\delta_l - p_l)\mu_{l,N}(x)$, with $l \leq i$, $\mu_{l,N}(x) = \langle z_l, \nabla_{y_l} y_l^j(x, \mathbf{1})\rangle$, and $Var_\delta[X_{l,N}(x)] = p_l(1 - p_l)\mu_{l,N}(x)^2$, assume that:*

- *There exists $a \in (0, \frac{1}{2})$ such that for all N and $l \in N$, $p_l \in (a, 1 - a)$,*

- *$lim_{N \to \infty} \frac{max_{k \in N}\mu_{k,N}(x)}{\sum_{l \in N}\mu_{l,N}(x)} = 0$, i.e. no single layer dominates in the computation,*

- *$v_{\frac{l}{N},\infty}(x) = lim_{N \to \infty} \frac{\sum_{l=1}^{N} Var_\delta[X_{l,N}(x)]}{N}$ exists and is finite.*

*Then, as $lim_{N \to \infty} y_l^j(x, \delta) \sim y_l^j(x, p) + \mathcal{N}(0, \frac{l}{N} v_{\frac{l}{N},\infty}(x))$*

The second assumption holds in practice, as otherwise it would imply a high degree of possible pruning of layers. Additionally, Hayou & Ayed (2021) demonstrate that it holds for ResNet.

If it is shown that $lim_{N \to \infty} \frac{1}{\sqrt{N}} \sum X_{l,N}^2(x) \to \mathcal{N}(0, \frac{l}{N} v_{\frac{l}{N},\infty}(x))$, then it implies that the last term mimics input-dependent gaussian noise.

In Theorem 3 and Lemma A4 of Hayou & Ayed (2021), it is shown that $lim_{n \to \infty} \frac{1}{s_n} \sum (X_{n,i} - \mu_{n,i}) = \mathcal{N}(0, 1)$ and $lim_{n \to \infty} \frac{1}{s_n^2} \sum \mathbb{E}[(X_{l,N})^2 1_{\{|X_{l,N}| > \epsilon s_n\}}] = 0$.

Using these results, it can be seen that $lim_{N \to \infty} \sum X_{l,N}^2(x) \to \mathcal{N}(0, 1)$, which leads to $lim_{N \to \infty} \frac{1}{\sqrt{N}} \sum X_{l,N}^2(x) \to \mathcal{N}(0, \frac{l}{N} v_{\frac{l}{N},\infty}(x))$.

The first half of the equation is equivalent to (as per Hayou & Ayed (2021)): $y_i^j(x, 1) + X_{l,N}^1(x) = y_i^j(x, 1) + (p_l - 1)\mu_{l,N}(x) \sim y_i^j(x, \mathbf{p})$ i.e. the average resulting network of training with skipping.

## G  FURTHER EXPERIMENTAL RESULTS

### G.1  TRAINING STABILIZATION

Here we study how vanilla SkipPipe (without occasional full-model executions) affect the convergence. We study this on LLaMa 500M trained for 35k steps in 4 different settings - full model execution, every

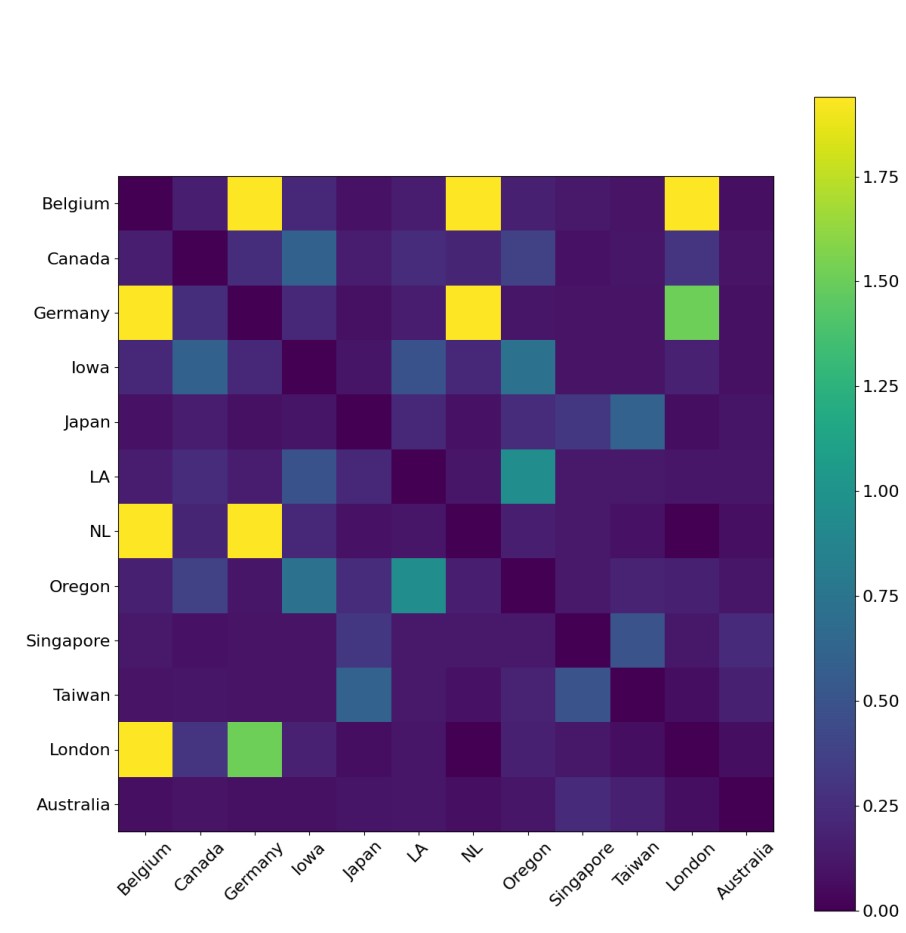

Figure 6: Bandwidths between the 12 different Google cluster locations

Table 4: Finetuned model evaluation results. For all higher is better. All results represent percentage of correct questions solved

| Task | Baseline | SkipPipe 25% |
| --- | --- | --- |
| BoolQ ↑ | 65.7% | **66%** |
| HellaSwag ↑ | **55.4%** | 52.6% |
| OpenBookQA ↑ | **36.2%** | 33.8% |
| ARC-easy ↑ | **63.3%** | 56.4% |
| GSM8k ↑ | **9.7%** | 5.9% |

5th step doing a full model execution otherwise skipping 33% of stages, every 9th step performing a full model, and a vanilla SkipPipe schedule, where no full model executions are performed. We see the results of this experiment in Fig. 7. There is a noticeable gap in perplexity between the vanilla schedule and the full model one. However, performing a full model run every 9th step drastically diminishes this gap. We observe no added benefit if we perform this every 4th step.

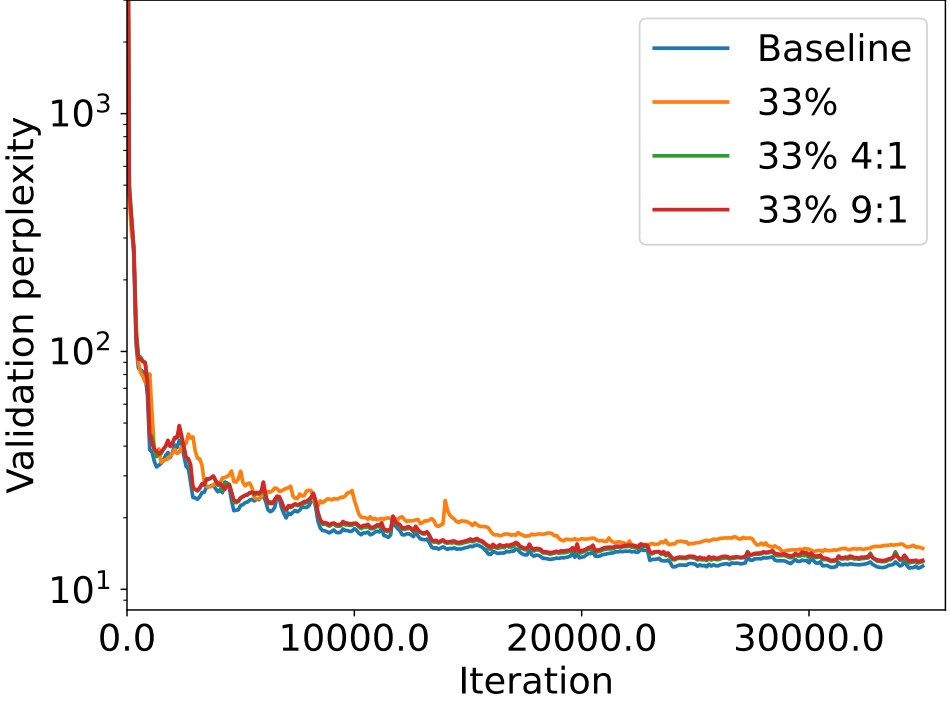

Figure 7: Comparison of different schedules for training and their effect on convergence.

## G.2 FINETUNED MODELS EVALUATION

We evaluate the two finetuned LLaMa 1B models on several common evaluation benchmarks. We make use of multiple choice ones: HellaSwag, ARC-easy, BoolQ, OpenBookQA; and one open ended: GSM8k. The results are presented in Table 4. While here the finetuned model with SkipPipe has a noticeable drop in performance, this is partially due to the fact that this was a model pretrained *without* skips, ergo the first few hundred iterations of finetuning were primarily spent learning these shorter paths.

### G.3 INFERENCE PERFORMANCE OF LLAMA-7B

Here we reaffirm our findings from Section 3.1 in the context of Large Language Models used in practice. While training billion parameter models is too expensive, here we focus on the inference case to confirm some of our previous findings. To such an end, we conduct an empirical performance study on skipping layers during inference on training a LLaMa-7B model Touvron et al. (2023), on the WikiPedia dataset Foundation. We consider four layer skipping strategies: (i) 0% skipping running the entire model end to end, (ii) 25% random skipping, (iii) 50% of random skipping, and (iv) 0% skipping and swapping the order of two chunks of size 4. We also repeat these four strategies by fixing the first four layer (they never get skipped or swapped). We summarize their loss in figure Fig. 8. Additionally, we demonstrate in the same setting the effect on inference of skipping any arbitrary stage in the LLaMa-7B model Touvron et al. (2023) during inference in 9.

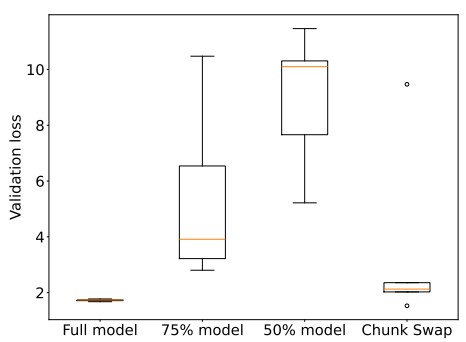
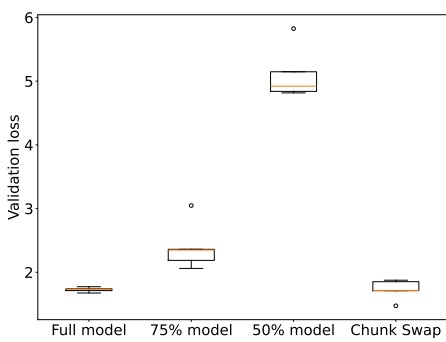

(a) Fully random skipping.  (b) Fixed first four layers.

Figure 8: The validation loss of LLaMa-7B under % of random skipping in pipeline training.

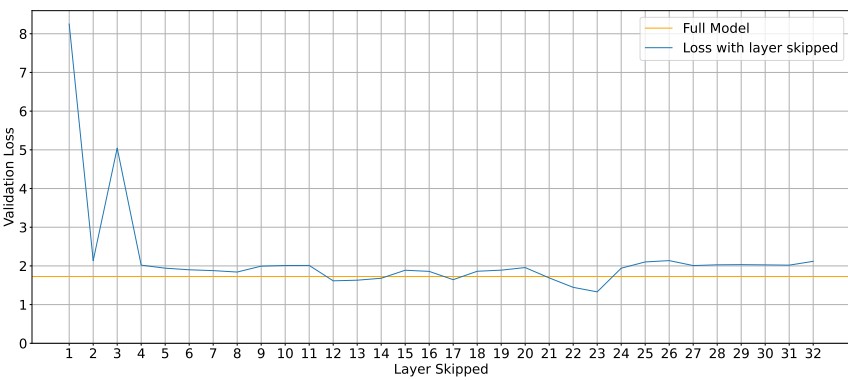

Figure 9: Validation loss when a given layer is skipped.