# OpenReview forum: "SkipPipe: Partial and Reordered Pipelining Framework for Training LLMs in Heterogeneous Networks"
_ICLR.cc/2026/Conference — ICLR 2026 Conference Withdrawn Submission_

### Official Review · Reviewer_DgC3 · 2025-10-24

**Soundness:** 2
**Presentation:** 3
**Contribution:** 2
**Rating:** 2
**Confidence:** 2

**Summary:**

A new pipeline parallel training method for distributed LLM training is introduced, where layers now allowed to be swapped and micro-batches can skip some fraction of layers. This results in better training throughput, which can also transfer to inference settings.

The authors evaluate simulated and real training settings and show that models which have been trained using the layer skipping method are more robust to layer skipping during inference.

**Strengths:**

- the proposed method has impressive results, reporting up to 250% speedup over no layer skipping
- only small drops in perplexity compared to no layer skipping are reported in experimental evaluations
- as a byproduct of training models using the SkipPipe algorithm, they can also be sped up during inference using layer skipping

**Weaknesses:**

W1. The paper propose a method for pipeline parallel training, which is sued for training large models. However, all experiments use quite small models, where such a parallelism method would never be used in practice. Unfortunately, this severely limits the strength of this publication. From a computational standpoint, running larger models should not be that expensive, since you only need to run a small amount of iterations to compute the throughput metrics. We need to be able to have some quantification of how the proposed method scales to larger models which is the setting where it would be applied.

W2. In the same vein, although more expensive to evaluate: we do not know whether the favorable convergence behavior remains as we scale beyond the 1B model range. Also, there are no benchmark evaluations and only perplexity is reported, which is not always indicative of performance on actual tasks.

W3. While the real deployment speedups are impressive (250%!), these stem from a "ultra-distributed" setting across continents. There do not seem to be real deployment evaluations in a setting within a single datacenter.

**Questions:**

Q1. In Figure 4b, using 25% skipping seems to lead to a significantly smoother validation loss curve than no skipping. Do you have an intuition as to why this might be the case (and is not the case in Fig 4a)?

Q2. The abstract and conclusion state an "up to 50%" speedup but in your experiments your report up to 250%. Is this a typo?

---

> ### Author Response · Authors · 2025-11-14
> **Reply to Reviewer DgC3 for Submission11835**
>
> We would like to thank the reviewer for their comments and issues raised, as they help us clarify the ideas in our paper. Below we address the issues they raised.
>
> **W1** *All experiments are on small models*
> Unfortunately, due to computational and memory constraints, large models above 1.5B are not possible to be trained, even for a few iterations. We refer to the analysis in Section 4.1, where a LLaMa-1.5B is trained and the batch size is increased (equivalent to sending larger activation size which correlates also with larger model size). We can observe that as the batch size is bigger (communication is larger in size), SkipPipe's improvement over other approaches becomes more apparent.
>
> [https://anonymous.4open.science/r/skippipe-43B2/upload-30035fc0e548f474375e50f1d1d112f0.png](https://anonymous.4open.science/r/skippipe-43B2/upload-30035fc0e548f474375e50f1d1d112f0.png)
>
> **W2** *On downstream evaluation*
>
> Below we have included performance (at 100% inference) of the base 500M models (without any SFT):
>
> |            |0%      |25%      |33%     |
> |------------|--------|---------|--------|
> |HellaSwag   | 34.8%  | 39.3%   | 37.9%  |
> |BoolQ       | 58.2%  | 56.7%   | 56.3%  |
> |OpenBookQA  | 27.6%  | 30%     | 27.8%  |
> |ARC Easy    | 38.3%  | 37.8%   | 37.5%  |
>
> As we can observe, similar to the loss lines and perplexity results, there is a negligible drop in performance between SkipPipe and sequential training. In fact, on two of the tasks (HellaSwag and OpenBookQA), the SkipPipe models are able to outperform sequential training.
>
> **W3 & Q2** *Real Deployment Results*
>
> Our real deployment results are specifically executed for a geo-distributed setting, rather than a single data center case, since our paper focuses on the heterogeneous network setting.
>
> Indeed, we observe the speed-up of 250%. One of the reasons we did not emphasize those results in the abstract is because we believe there is a significant fluctuation happening in the real deployment, and it could lead to an unfair advantage for us. Moreover, we want to emphasize the simulated experiments, as they can be reproduced by third parties as well.
>
> **Q1** *25% skipping seems to lead to a significantly smoother validation loss curve than no skipping*
>
> Indeed, that is an interesting observation. We do not have a specific intuition on why the loss function seems smoother.

---

> > ### Author Response · Authors · 2025-11-18
> > **Edit notification**
> >
> > As the previous image hosting platform was having some issues, we updated the image links in our original reply to be more easily accessible.

---

> > > ### Comment · Reviewer_DgC3 · 2025-11-24
> > >
> > > Thank you for your response.
> > >
> > > - Overall, W1 remains unaddressed and prevents me from raising my score. Compute constraints preventing the running of significantly larger models than 1.5B-scale are not convincing to me, given that in the paper H100 nodes are already used and should easily accommodate larger models. Pipeline parallelism is in practice only used with much larger models (e.g. >=13B), the small-scale experiments in the paper unfortunately are too much of a mismatch compared to any real application of the proposed method. Equating larger batch sizes to larger models is unfortunately not correct, as the two settings have different compute and communication patterns. The paper would significantly benefit from showing evaluations at a scale where the proposed method would actually be used.
> > >
> > > - Thank you for providing the downstream task performance evaluations of the 500M model. Although the scores are quite close to random performance (e.g. BoolQ or OpenBookQA), this partially addresses the lack of downstream task evaluations. However the point about scaling behavior of the lack of performance drop due to layer skipping to larger models remains unexplored.
> > >
> > > - Thank you for addressing W3&Q1&Q2.

---

### Official Review · Reviewer_wpwm · 2025-10-29

**Soundness:** 2
**Presentation:** 3
**Contribution:** 2
**Rating:** 2
**Confidence:** 3

**Summary:**

This paper proposes SkipPipe, a novel framework to randomly skip some stages of PP for LLM training. The core contribution is two-fold:
- With both theoretical and empirical studies, the authors show that skipping a proportion of LLM layers has negligible impact on the loss convergence.
- Skipping these layers results in end to end training accelerations. Evaluations show up to 50% acceleration with no significant accuracy degrade.

**Strengths:**

- A novel design on the path finding and scheduling algorithm to avoid collisions.
- Evaluation shows significant training acceleration via both simulation and geo-distributed experiments.
- Evaluation on convergence shows similar training loss curves compared to baseline. Moreover, during inference it shows robustness against layer dropping.

**Weaknesses:**

Major Concern:
The primary concern is with the methodology of pipeline stage skipping. While there is literature on similar topics, existing work does not fully support the authors’ claim of negligible accuracy loss; in some cases, it suggests the opposite. For example, LayerSkip [1] consistently frames early exits as a trade-off between computation and accuracy. Figure 5 in LayerSkip shows significant accuracy degradation during continual pretraining when early exiting is applied. To address this, LayerSkip proposes self-speculative decoding to correct the accuracy. In line with this, I concur that optimizations like SkipPipe should be treated as a trade-off between compute efficiency and model performance, rather than claiming negligible impact on accuracy. At minimum, additional evidence should be provided to support the claim that LLMs are robust to SkipPipe-style stage skipping, which is not convincingly demonstrated in the current paper.

Other Concerns:
- Limited evaluation and scale: The experiments are conducted on a 0.5B model for pretraining and a 1B model for fine-tuning. It remains unclear how SkipPipe affects convergence and performance on larger models with more layers or stages.
- Performance degradation on downstream tasks: Table 4 shows noticeable accuracy drops across most tasks, which appears to contradict the claim that SkipPipe does not affect accuracy.
- Theoretical analysis limitations: The theoretical discussion focuses on individual layers, whereas SkipPipe skips entire stages composed of consecutive layers. The analysis may therefore not fully capture the impact of the proposed method.

Suggestions:
I recommend the authors reframe SkipPipe as an approach that enables an effective trade-off between accuracy and latency, rather than claiming negligible accuracy loss. It is counterintuitive that skipping layers would fully preserve accuracy. The paper could be strengthened by demonstrating that SkipPipe achieves better efficiency compared to alternative methods with the same goal, such as model pruning, quantization, or direct model size reduction, especially in geo-distributed setups.

Reference:
[1] LayerSkip: Enabling Early Exit Inference and Self-Speculative Decoding, ACL 2024. https://aclanthology.org/2024.acl-long.681.pdf

**Questions:**

- Is the theoretical analysis still correct considering SkipPipe is dropping consecutive layers?
- Is there a significant overhead for the path scheduling algorithm to run?
- There might be some confusion on the evaluation setup, the authors explains the 250% acceleration against DT-FM is caused by improved network compared to DT-FM experiments. Can authors clarify that whether the network and hardware setup are the same for SkipPipe and DT-FM?

---

> ### Author Response · Authors · 2025-11-14
> **Reply to Reviewer wpwm for Submission11835**
>
> We would like to thank the reviewer for their comments and issues raised, as they help us clarify the ideas in our paper. Below we address the issues they raised.
>
> **W1** *Limited evaluation and scale*
>
> Unfortunately, due to computational and memory constraints, large models above 1.5B are not possible to be trained. We refer to the analysis in Section 4.1, where a LLaMa-1.5B is trained and the batch size is increased (equivalent to sending larger activation size which correlates also with larger model size). We can observe that as the batch size is bigger (communication is larger in size), SkipPipe's improvement to other approaches becomes more apparent.
>
> [https://anonymous.4open.science/r/skippipe-43B2/upload-30035fc0e548f474375e50f1d1d112f0.png](https://anonymous.4open.science/r/skippipe-43B2/upload-30035fc0e548f474375e50f1d1d112f0.png)
>
> **W2** *Performance degradation on downstream tasks*:
>
> We would like to point out that the results of Table 4 are on SFT-only training. This drop in performance is in line with our training performance in Figure 4b. From Appendix G.2: "While here the finetuned model with SkipPipe has a noticeable drop in performance, this is partially due to the fact that this was a model pretrained without skips, ergo the first few hundred iterations of finetuning were primarily spent learning these shorter paths".
> Below we have included performance (at 100% inference) of the base 500M models (without any SFT):
>
> |            |0%      |25%      |33%     |
> |------------|--------|---------|--------|
> |HellaSwag   | 34.8%  | 39.3%   | 37.9%  |
> |BoolQ       | 58.2%  | 56.7%   | 56.3%  |
> |OpenBookQA  | 27.6%  | 30%     | 27.8%  |
> |ARC Easy    | 38.3%  | 37.8%   | 37.5%  |
>
> As we can observe, similar to the loss lines and perplexity results, there is a negligible drop in performance between SkipPipe and sequential training. In fact, on two of the tasks (HellaSwag and OpenBookQA), the SkipPipe models are able to outperform sequential training.
>
>
> **W3 & Q1** *Is the theoretical analysis still correct considering SkipPipe is dropping consecutive layers?*
>
> The theoretical analysis focuses on individual layers skipped. It makes no assumptions on whether the skipped layers are consecutive or not. As such the analysis is equally valid for skipping k individual layers at random locations or skipping a group of k consecutive layers (which is what a stage is). Thus, the theoretical analysis is still correct.
>
> **Q2** *Is there significant overhead to running the scheduling*
>
> The scheduling algorithm is computationally expensive, however with our proposed heuristics, it runs within less than 10 minutes. Additionally, compared to the entire time for training an LLM, such an overhead is negligible.
>
> **Q3** *Can authors clarify that whether the network and hardware setup are the same for SkipPipe and DT-FM?*
>
> Yes, for our experiments, both the baseline and SkipPipe use the exact same hardware, bandwidth and latency matrices across all experiments. We perform two types of experiments - simulated on several nodes with bandwidth and latency matrices taken from the DT-FM paper and a real-world deployment over several Google Cloud instances. What we meant by "We attribute this greater speed up due to the much higher variability in bandwidth between locations relative to the ones used in DT-FM" is that the real-world deployment over Google Cloud instances probably had much more varied matrices than those on AWS (as used in the DT-FM paper). Thus, this greater variability is the reason for the much stronger speedup of SkipPipe over the baseline relative to the simulated experiments.

---

> > ### Author Response · Authors · 2025-11-18
> > **Edit notification**
> >
> > As the previous image hosting platform was having some issues, we updated the image links in our original reply to be more easily accessible.

---

### Official Review · Reviewer_yUVr · 2025-10-31

**Soundness:** 3
**Presentation:** 3
**Contribution:** 3
**Rating:** 6
**Confidence:** 3

**Summary:**

This paper presents an algorithm to optimize the training of an LLM--here synonym with decoder transformer--over distributed nodes, given the bandwidth and latency between any of them.

Five constraints are identified related for three of them to the tolerance of a decoder transformer to change in the order of evaluation of its layers, and for two to the memory constraints and the necessity to avoid collision. The dispatching of minimatches and stages reduces to a  discrete (and complex) discrete optimization problem.

Experiments demonstrate excellent performance compared to baselines.

**Strengths:**

The problem addressed is very important, as it could unlock large-scale collaborative training and more generally allows a better use of existing infra. The method is well justified and simple in the statement of the optimization problem, but involve some technicalities for the optimization. The performance seems excellent compared to baselines, i particular the results with a "real" setup of Appendix E are very reassuring. The writing is clear (except the explanation related to TC2).

**Weaknesses:**

The methods is restricted to a specific, although currently super dominant, class of models. The experiments are unfortunately mostly simulation, and far from large scale, which is quite problematic for such a class of methods. Performance are not given in tangible quantities (e.g. wall clock times), it may be clear to people in that precise field, but it would then be helpful to have even rough estimates.

**Questions:**

1. I presume speed measurement do not need more than a few percent of a full training, why no experiments with a 8B or bigger? How does this method combines with e.g. FSDP ?

2. Please re-explain TC2, which seems to be a heuristic more than a constraint.

3. There seems to be no substantial degradation in Fig 4 (a), what happens with e.g. 50%, 75%, 90% ?

4. Is DT-FM the best available alternative? How does this compare with a standard cluster?

---

> ### Author Response · Authors · 2025-11-14
> **Reply to Reviewer yUVr for Submission11835**
>
> We would like to thank the reviewer for their comments and issues raised, as they help us clarify the ideas in our paper. Below we address the issues they raised.
>
> We are currently running the following experiment, and we will share the results as soon as possible:
> - Running SkipPipe training with 50% skip ratio
>
>
> **W1** *Experiments are primarily simulated*
>
> We provide experiments in a real geo-distributed setting across several continents. We focus on the simulated experiments as they are easier to reproduce and allow for a more controlled environment.
>
> **Q1** *On larger training and FSDP*
>
> Unfortunately, due to computational and memory constraints, large models above 1.5B are not possible to be trained, even for a few iterations. We refer to the analysis in section 4.1, where a LLaMa-1.5B is trained and the batch size is increased (equivalent to sending larger activation size which correlates also with larger model size). We can observe that as the batch size is bigger (communication is larger in size), SkipPipe's improvement to other approaches becomes more apparent.
>
> [https://anonymous.4open.science/r/skippipe-43B2/upload-30035fc0e548f474375e50f1d1d112f0.png](https://anonymous.4open.science/r/skippipe-43B2/upload-30035fc0e548f474375e50f1d1d112f0.png)
>
>
> **Q2** *On TC2*
>
> As hardware has the physical limitation of processing one microbatch at a time, if a device receives two microbatches at once, one will be slowed down by the processing time of the other. Thus, in an execution order of mb1 1-3-4 and of mb2 2-3-4, node 3 would receive two microbatches at once (assuming 1 time unit of communication and processing time). Thus one microbatch (assume mb2) would have a processing time of 6 time units. If its order is swapped to 2-4-3, it will have a processing time of 5 time units, the same as mb1. This motivates us to reduce collisions as much as possible via the Multi Agent Path Finding algorithm. As we observe, reducing collisions improves throughput by 10%. If the scheduler is not aware of such a constraint, it can schedule unknowingly several microbatches at the same time on a node (as it assumes they can pass through together), potentially increasing the processing time. Typical sequential execution pipelines do not suffer from this problem, as a node can only receive from one other node (the one in the previous stage).
>
> **Q3** *How will performance be affected at 50%?*
>
> This is an interesting direction to observe. We observe a general trend in the decrease of performance as the skip ratio is increased.
> We are currently running additional experiments on the performance of SkipPipe at 50% skip rate and we will share the results here as soon as possible.
>
> **Q4.1** *Is DT-FM the only comparable baseline?*
>
> There exist other scheduling solutions such as Metis [1] and HexiScale [2], but these predominantly focus on heterogeneous clients. In our experiments, we utilize homogeneous clients where each has the same computational and memory capacity. However, our solution is compatible with a heterogeneous client setting, requiring only the node to stage mapping being replaced by one of these heterogeneous algorithms.
>
> **Q4.2** *How does this compare with a standard cluster?*
>
> Our solution targets geo-distributed training. It is possible to apply SkipPipe to a single data center, however the benefit there would primarily be in saved computation times due to stage skipping.
>
> [1] Taegeon Um, Byungsoo Oh, Minyoung Kang, Woo-Yeon Lee, Goeun Kim, Dongseob Kim, Youngtaek Kim, and Mohd Muzzammil. Metis: Fast Automatic Distributed Training on Heterogeneous GPUs. 2024
>
> [2] Ran Yan, Youhe Jiang, Xiaonan Nie, Fangcheng Fu, Bin Cui, Binhang Yuan.
> HexiScale: Accommodating Large Language Model Training over Heterogeneous Environment. 2024

---

> > ### Author Response · Authors · 2025-11-18
> > **Edit notification**
> >
> > As the previous image hosting platform was having some issues, we updated the image links in our original reply to be more easily accessible.

---

### Official Review · Reviewer_g3rd · 2025-10-31

**Soundness:** 3
**Presentation:** 2
**Contribution:** 2
**Rating:** 4
**Confidence:** 4

**Summary:**

This work proposes a new pipeline-parallel approach to pre-training large language models, SkipPipe, grounded in the empirical robustness of LLMs to layer skipping and layer reordering. SkipPipe can intelligently schedule data paths at the micro-batch level to boost the efficiency of LLM pre-training. The results show robust performance and time savings compared to the full pipeline approach. The authors also observe robustness to layer omission during inference for models trained with SkipPipe.

**Strengths:**

- S1. This work tackles a critical and expanding area of ML: efficient LLM pre-training on geo-distributed networks.
- S2. The proposed method, SkipPipe, is based on established observations regarding the robustness of LLM to specific perturbations, i.e., layer skipping and layer re-ordering. Leveraging the known robustness of LLMs to skipped or shuffled layers is a clever idea that underpins SkipPipe’s design.
- S3. Despite being based on established techniques, SkipPipe has the merit of coupling them in a principled way, driven by constraints and scheduling algorithms, producing predictable savings and emergent robustness.

**Weaknesses:**

- W1. The overall training speedup due to SkipPipe’s scheduling algorithm, beyond existing techniques that also skip layers, appears modest. While SkipPipe achieves up to 50% speedup over a standard pipeline, a large portion of this gain comes simply from skipping layers rather than the new scheduling; the novel contribution yields only an incremental improvement (~10–20% beyond a baseline that already skips layers).
- W2. The experimental evaluation is incomplete in terms of demonstrating the model’s effectiveness. The paper evaluates training speed and shows validation perplexity, but it lacks downstream task evaluations (e.g., in-context learning tasks or other language understanding benchmarks) to show the impact of SkipPipe on actual performance. Including such experiments would strengthen the paper.
- W3. The contribution has limited novelty. Similar ideas of skipping layers or reordering in training have been explored (e.g., layer drop for skipping layers during training, or prior heterogeneous scheduling methods for pipeline arrangement). SkipPipe’s main innovation is integrating these existing ideas (skip training and path scheduling) – a useful engineering improvement, but arguably an incremental advance rather than a fundamentally new concept.
- W4. The paper’s claims about ‘robust inference’ (tolerating dropped layers) are not well contextualized relative to prior work. Prior research (e.g., dropout, layer drop) shows that training with missing parts can improve robustness. The authors should clarify whether this phenomenon is unique to SkipPipe or simply an expected outcome of partial training (and compare to a baseline model trained normally: how much worse would a standard-trained model perform if some layers are skipped at inference?). Currently, the robustness result is presented without discussing such context or baselines, leaving it unclear how novel or significant this finding is
- W5. The authors provided an anonymous code repository link, but the code is currently unavailable (the repo is empty/inaccessible at review time), hindering reproducibility.
- W6. The writing could be clearer in places. Some sections (e.g., the theoretical constraints or algorithm description) were hard to follow due to dense notation and could be better explained. Improving the clarity and presentation (figures, notation) would help readers understand the contributions.

**Questions:**

- Q1. Can the authors discuss the limitations of assuming a static (B, Lambda) graph and how their method would change if such a graph is dynamic (changing over time)?
- Q2. The evaluation of model quality is very limited. Can the authors add sensible ICL downstream tasks to their evaluation instead of the limitedly insightful validation perplexity on some benchmarks?
- Q3. Most time savings compared to the baseline seem to come from skipping layers. Can the authors quantify how much communication (in bytes) DT-FM-skip and SkipPipe are saving with respect to the baseline DT-FM? How strong is the correlation between such savings in bytes and the savings in time? This comparison could help readers understand SkipPipe’s contributions.

---

> ### Author Response · Authors · 2025-11-14
> **Reply to Reviewer g3rd for Submission11835**
>
> We would like to thank the reviewer for their comments and issues raised, as they help us clarify the ideas in our paper. Below we address the issues they raised.
>
> **W1 & W3** *Novelty and incremental improvement*
>
> Indeed, there have been works on both layer skipping and scheduling for heterogeneous networks. Integrating these existing ideas is reflected in DT-FM-skip in our experiments. To be precise, even the results we present for DT-FM-skip also includes some additional improvements (see Appendix B.1) rather than naive integration of the existing ideas. In our experiments, it can be seen that SkipPipe outperforms DT-FM-skip. Considering pre-training of LLMs being quite expensive, 10-20% improvements can reduce the cost of training in the order of tens-to-hundreds of thousands of dollars.
>
> The novelty of SkipPipe comes from the optimisation of the skipping in a heterogeneous setting. For that, we first formalise the problem (specifically collusions - that did not exist in previous settings) and introduce several constraints on the optimal path selection. We then use our Algorithm 1 given in Appendix C to solve this problem.
>
> **W2 & Q2** *Donwstream evaluation*
>
> Here, we have included performance (at 100% inference) of the base 500M models (before SFT):
>
> |            |0%      |25%      |33%     |
> |------------|--------|---------|--------|
> |HellaSwag   | 34.8%  | 39.3%   | 37.9%  |
> |BoolQ       | 58.2%  | 56.7%   | 56.3%  |
> |OpenBookQA  | 27.6%  | 30%     | 27.8%  |
> |ARC Easy    | 38.3%  | 37.8%   | 37.5%  |
>
> As we can observe, similar to the loss lines and perplexity results, there is a negligible drop in performance between SkipPipe and sequential training. In fact, on two of the tasks (HellaSwag and OpenBookQA), the models trained with SkipPipe outperform the baseline (sequential training).
>
> **W4** *Is the resistance to pruning during inference unique to SkipPipe*
>
> Indeed, this is not a unique effect, but an added benefit of such training. Additionally, unlike methods like LayerSkip, which allow for skipping the last k stages during inference, SkipPipe can tolerate dropping any random k stages. Thus, if such models are deployed in a decentralized setting, stages in the middle can crash without significantly affecting the performance.
>
> **W5** *Repo being unavailable*
>
> We thank the reviewer for catching the fact that our anonymous repository has expired. We have remedied this issue and it should now be fully accessible!
>
> **W6**  *Clarity and presentation of the paper*
>
> We thank the reviewer for the comment. We are working on improving the writing of the paper, and clarifying the technical details and notations used in the paper.
>
> **Q1** *Assumption on static graph*
>
> Indeed, we assume static graphs, which is also commonly used in previous works. Dynamic scheduling algorithms require self-stabilizing approaches [1]. Our approach would require frequent recomputations of the microbatch scheduling (multi-agent path finding problem), as redistributing nodes in stages would be too costly to be performed ad-hoc. To the best of our knowledge, there is no algorithm for multi-agent path finding in a changing graph. We will amend the final version of the paper with the analysis above.
>
>
> **Q3** *Communication reduction for SkipPipe in bytes*
>
> We present our analysis assuming no compression (or minimal lossless compression in practice) of communication is applied. Let us consider a simple example of a 6-stage setup (0-1-2-3-4-5). A traditional training algorithm would need to perform 12 inter-node communications (6 forward and 6 backward, since stage 0, the data holder, needs to be both at the beginning and at the end to compute the loss). SkipPipe at 33% skip rate needs to perform only 8 (4 forward and 4 backward). As the activations and gradients are of the same size between the two settings, the reduction in communication in bytes is the skip rate chosen per microbatch. SkipPipe has also the additional benefit of lower memory requirment (again equivalent to the skip ratio), thus allowing for more efficient communication and device usage. These two factors contribute to the speed ups observed in the experimental section
>
>
> [1] Nikolay Blagoev, Bart Cox, Jérémie Decouchant, Lydia Y. Chen. Go With The Flow: Churn-Tolerant Decentralized Training of Large Language Models. 2025

---

### Author Response · Authors · 2025-11-18
**Main comment**

We thank all reviewers for their valuable feedback. We added new tables to strengthen the paper. All updates will be reflected in the final version.

**Additional experiments**
- **Downstream evaluation** - all reviewers requested downstream evaluation of the 500M pretrained models. This has now been added (on the models without an additional SFT step) and we observe, similar to the perplexity evaluation, little to no drop in performance at 100% inference (i.e. no skipping during evaluation).
- **Higher skip rates** - reviewer *yUVr* requested results on convergence of the 500M models with 50% skip rate. This is currently being ran and once the training has finished, we will share the results here.

We hope these additional experiments and results address the concerns of the reviewers


*Note: two reviewers (*DgC3*, *wpwm*) expressed the need for convergence results on larger models. This however is prohibitively expensive - pretraining models from scratch is very resource, time, and memory demanding. We hope that the current models trained and the added theory are sufficient to demonstrate the viability of SkipPipe*

---

### Note · Authors · 2026-01-14

**Comment:**

We are withdrawing with the aim to improve for future resubmissions. We thank the reviewers and the AC for their efforts in this turbulent submission cycle.

**Withdrawal Confirmation:**

I have read and agree with the venue's withdrawal policy on behalf of myself and my co-authors.